# Orchid Micropropagation Using Conventional Semi-Solid and Temporary Immersion Systems: A Review

**DOI:** 10.3390/plants12051136

**Published:** 2023-03-02

**Authors:** Potshangbam Nongdam, David G. Beleski, Leimapokpam Tikendra, Abhijit Dey, Vanlalrinchhani Varte, Soumaya EL Merzougui, Vania M. Pereira, Patricia R. Barros, Wagner A. Vendrame

**Affiliations:** 1Department of Biotechnology, Manipur University, Canchipur 795003, India; 2Environmental Horticulture Department, University of Florida, Gainesville, FL 32611, USA; 3Department of Life Sciences, Presidency University, Kolkata 700073, India; 4Department of Molecular Reproduction, Development and Genetics, Indian Institute of Science, Bangalore 560012, India; 5Laboratory of Biotechnologies and Valorization of Natural Resources, Department of Biology, Faculty of Sciences, Ibn Zohr University, Agadir 80000, Morocco; 6Department of Soil, Federal University of Vicosa, Vicosa 36570-900, Brazil

**Keywords:** orchids, micropropagation, explants, semi-solid media, temporary immersion system, conservation

## Abstract

Orchids, with their astonishingly stunning flowers, dominate the international floricultural market. They are considered prized assets for commercial applications in pharmaceutical and floricultural industries as they possess high therapeutic properties and superior ornamental values. The alarming depletion of orchid resources due to excessive unregulated commercial collection and mass habitat destruction makes orchid conservation measures an extreme priority. Conventional propagation methods cannot produce adequate number of orchids, which would meet the requirement of these ornamental plants for commercial and conservational purposes. In vitro orchid propagation using semi-solid media offers an outstanding prospect of rapidly producing quality plants on a large scale. However, the semi-solid (SS) system has shortcomings with low multiplication rates and high production costs. Orchid micropropagation using a temporary immersion system (TIS) overcomes the limitations of the SS system by reducing production costs and making scaleup and full automation possible for mass plant production. The current review highlights different aspects of in vitro orchid propagation using SS and TIS and their benefits and drawbacks on rapid plant generation.

## 1. Introduction

Orchids are amazingly stunning ornamental plants belonging to the Orchidaceae family, comprising 30,000–35,000 species, numerous hybrids, and varieties [1,2,3]. Most orchids carry a high price in the international floriculture market as they have great floricultural appeal with their diversely colored attractive flowers having varied fragrances, shapes, and sizes [4,5,6]. Orchids also possess good therapeutic properties despite their outstanding ornamental values due to the rich contents of beneficial phytochemicals [7,8]. The therapeutic applications of orchids are found in various traditional medicinal systems, including Ayurveda [9,10]. The high ornamental and therapeutical values of orchids make them a prized asset for commercial application in the floriculture and pharmaceutical industries [11]. However, the orchid population witnessed a profound reduction in their natural habitats primarily due to unregulated commercial collection, deforestation, and massive habitat destruction [12]. The alarming orchid population decline warranted the whole family to be covered under Appendix-II of the Convention on International Trade in Endangered Species of Wild Fauna and Flora (CITES) [13,14]. They are also among the most threatened taxonomic groups [15], with more than 600 orchid species enlisted as threatened in the Red List of the International Union for the Conservation of Nature [16]. Commercial orchid production through conventional propagation is not always effective as it is slow, time-consuming, and labor-intensive [2]. Furthermore, propagation through the seed culture method has serious drawbacks as the numerous light tiny exalbuminous seeds have a very low germination rate (0.2 to 0.3%) in nature due to the lack of functional endosperms and the necessity of suitable mycorrhizal fungal association [17,18,19]. The shortcomings of conventional orchid propagation methods can be prevailed by using plant tissue culture techniques to produce plants rapidly on a commercial scale. Several orchids have been micropropagated successfully on the semi-solid (SS) media supplemented with various plant growth regulator combinations and concentrations using different explants [20,21,22,23,24,25]. However, the traditional SS culture system for orchid propagation is filled with limitations of low multiplication rate, high production cost, and the associated problem of stem and root hyperhydricity [26,27]. Application of a temporary immersion system (TIS) can significantly reduce production costs, making it economically viable by allowing scaleup and full automation and eliminating inherent plant physiological problems associated with SS and continuous liquid culture systems [28,29]. TIS has been employed to propagate orchid species using different media, growth regulators, and explants [30,31,32,33]. The present review paper emphasizes different aspects of in vitro orchid propagation, conventional SS and TIS, benefits and weaknesses, and their applications for efficient large-scale plant production.

## 2. Orchid Micropropagation Using Semi-Solid Media

The semi-solid culture (SS) system, which uses agar-gelled media, has been employed for the successful micropropagation of several orchids [34,35,36,37,38]. Micropropagation using a SS medium has several advantages over conventional orchid propagation methods, which are plagued with several limitations of being slow, labor-intensive, and highly time-consuming [39]. The techniques offer the prospect of multiplying genetically identical plants on a large scale quickly using small plant parts as explants without sacrificing the mother plants. Moreover, mass multiplication of disease-free plants can be performed in a small space independent of the seasonal cycle [21]. Transportation and export of small micropropagated plants are also convenient as they can be easily packed and have no quarantine problems as they are clean, healthy, and pathogen-free [39]. The success of orchid micropropagation depends mainly on the choice of appropriate culture media, as they provide essential nutrients for the in vitro growth of the plants [40]. Murashige and Skoog (MS) [41], Mitra (M) [42], Knudson C (KC) [43], and Vacin and Went (VC) [44] are widely used culture media for in vitro orchid propagation. The media contain macro and micronutrients, which are critical factors determining the success of in vitro orchid culture [45]. They differ in their composition of macro and micronutrients. Ammonium is present in higher amounts as ammonium nitrate in MS compared to KC and VW medium, where it exists as ammonium sulfate. VW and KC have more potassium phosphate monobasic content though the composition of potassium nitrate is lower than in MS medium [46]. MS and KC media have been extensively used for efficient seed germination and in vitro propagation of several orchids [47,48,49,50]. VC medium, on the other hand, has been mainly utilized for the proliferation of protocorm-like bodies, which helped in the orchid clonal multiplication on a large scale [51]. Modification of culture media is made by variation in the concentration of nutrient components and integrating organic additives like coconut water (CW) and potato extract to suit the requirement of a particular species. The addition of organic supplements in the culture media had a promotory effect on in vitro seed germination and culture growth in some orchids [52,53,54]. Incorporating plant growth regulators is vital for the success of orchid micropropagation as they enhance culture growth with cytokinins promoting shoot initiation, proliferation, and plant regeneration and auxins accelerating root induction and multiplication [55,56]. Plant growth regulators may improve tissue development but show adverse effects at particular elevated concentration. The embryo formation from leaf explants was delayed at high auxin concentration (3 mg L^−1^) in *Oncidium* ‘Gower Ramsey’ [57]. Tikendra et al. [58] also detected somaclonal variation in micropropagated *Dendrobium fimbriatum* propagated under high cytokinin concentration (4.8 mg L^−1^).

### 2.1. Micropropagation Employing Different Explants

Different explants, such as seeds, shoot, root and leaf apex, nodal segments, rhizome, pseudobulb, and inflorescences, have been used commonly to micropropagate different orchids [2,18,20,26,39,40]. The selection of the right explant type is critical as its response to cultural conditions is crucial for successful orchid micropropagation [59]. The juvenile state [60], size, position, orientation [61], and collection season [62] of the explants are vital for their responsive nature. Kaur and Bhutani [60] also stated the favorability of young tissues of explants over mature ones as they possess high regeneration capability. Explants established from in vitro-raised plants are preferred over those derived from in vivo plants as they are not contaminated and have more significant differentiation potential [56]. 

Orchid seeds germinate very poorly in nature due to their small, light, and exalbuminous seeds, which require symbiotic association with mycorrhizal fungi [63]. Ever since successful in vitro asymbiotic seed germination was performed on KC medium [64], there have been several reports on orchid micropropagation using seed explants [54,65,66,67,68,69,70]. The germinated seeds swell, forming round-shaped protocorms that differentiate into shoots and roots, subsequently developing into seedlings with leaves and roots (Figure 1). Srivastava et al. [71] reported the best germination (89.28 ± 3.42%) of *Aerides ringens* seeds on KC medium augmented with 4.44 µM 6-benzylaminopurine (BAP) and 500 mg L^−1^ peptone, while Rodrigues et al. [67] observed 100% seed germination of *Cyrtopodium saintlegerianum* on KC medium incorporated with 3 g L^−1^ activated charcoal (AC). Whereas Wu et al. [72] demonstrated high seed germination (93%) of *Renanthera imschootiana* on ¼ MS medium supplemented with 0.5 mg L^−1^ 1-naphthalene acetic acid (NAA), 20% CW, 1.0 g L^−1^ peptone, 10 g mL^−1^ sucrose, and 1.0 g mL^−1^ AC, the maximum seed germination response of *Caladenia latifolia* was witnessed on ½ MS with 5% CW. Nongdam and Chongtham [47] also noticed the highest seed germination of *Cymbidium dayanum* on M medium fortified with 0.7 mg L^−1^ NAA.

Wimber [73] was the first to use leaf segments successfully for in vitro orchid propagation. The leaf culture in monopodial orchids is beneficial as no mother plant is sacrificed, and the explants are available throughout the season [74]. Chookoh et al. [75] showed only the inner expanding leaves of *Tolumnia* Snow Fairy producing protocorm-like bodies (PLBs) on MS medium augmented with 4 mg L^−1^ BA and 0.5 mg L^−1^ NAA, while outer leaves showed no in vitro growth response. Pathak et al. [76] observed the maximum regeneration potential of the whole leaf segments (0.5–1 cm long) when inoculated on Mitra (M) medium fortified with 5 mg L^−1^ kinetin (KN) producing maximum plantlets (25). Decruse and Gangaprasad [77] found the in vitro seedling-derived leaves of *Smithsonia maculate* producing shoots prominently in Woody Plant Medium (WPM) augmented with 10 mg L^−1^ BAP and 1 mg L^−1^ indole-3-acetic acid (IAA). Manokari et al. [25] observed leaf explants of *Vanilla planifolia* inducing direct somatic embryogenesis (92.0%) on MS supplemented with 1.0 mg L^−1^ BAP, 1.0 mg L^−1^ KN, and 2.5 mg L^−1^ silver nitrate (SN) generating maximum somatic embryos (30.0 per explant).

Morel [78] employed apical meristem for the first time to propagate *Cymbidium* successfully. Since then, many orchids have been micropropagated using shoot tip explants [79,80,81,82]. Priyakumari et al. [83] found ½ MS medium enriched with 4 mg L^−1^ BA favorable for early bud break of shoot tip explants of *Dendrobium* Sonia “Earsakul” producing maximum shoots (4.66) on medium incorporated with 2.0 mg L^−1^ KN and 0.1 mg L^−1^ NAA. The bisected shoot tips of the *Phalaenopsis* hybrid were most suited for producing PLBs and tissue multiplication on MS medium supplemented with varying concentrations of thidiazuron (TDZ) [84]. Winarto and Samijan [85] witnessed maximum axillary shoot proliferation (7.0 shoots per explant, 1.0 cm shoot height, and 9.8 leaves per explant) on MS appended with 1.5 mg L^−1^ BAP and 0.25 mg L^−1^ NAA, while root production was higher on medium containing 150 mL L^−1^ CW. Pant and Thapa [86], while using the shoot tips of *Dendrobium primulinum*, obtained the highest rootless healthy shoots (4.5 shoots per culture) on MS medium augmented with 1.5 mg L^−1^ BAP, while the best root formation was achieved on medium with 5 mg L^−1^ IAA. 

Kerbauy [87] used root tips of *Catasetum* to generate PLBs directly without forming calli on MS medium. Organic additives such as bacto-peptone and AC enhanced PLBs formation though larger explant dimension reduced PLBs regeneration. Sharma [88] found the proximal region of the root segment of *Rhynchostylis retusa* involved in the highest plant regeneration (31%) on M medium fortified with 3 mg L^−1^ KN and 1 mg L^−1^ NAA, forming 28 plantlets in 15 weeks. Verma and Pathak [89] showed that 100% of the root explants of *Cymbidium aloifolium* responded to M medium with 3 mg L^−1^ NAA forming healthy plantlets via shoot bud formation in 55 days of culture. Picolotto et al. [90] found root explants of *Cyrtopodium paludicolum* differentiating proficiently into shoots and roots through intervening PLBs formation on KC medium supplemented with 1.34 μM NAA and 2.27 μM TDZ. However, the absence of NAA caused PLBs formation to fail, demonstrating the synergistic effect of both NAA and TDZ. A brief account of the recent works on orchid micropropagation using different explants in the last four years is presented in Table 1.

### 2.2. Limitations of SS and Liquid Culture

The SS system has been employed for the successful micropropagation of several orchids [12,18,19,20,25,34,48,61,69]. However, there are many limitations to using the SS system for traditional in vitro plant propagation. The production cost using this method is high with the necessity of buying a massive number of small to medium size culture vessels apart from more than 40–60% of the cost attributed to labor charges and loss of plants during the acclimatization due to stem and root hyperhydricity [26,91,92,93]. The method is highly labor-intensive, requiring cleaning and filling of many containers during media preparation for frequent plant subcultures to avoid nutrient exhaustion from continuous culture growth in small containers [94,95,96]. SS medium with a gelling agent also makes the scaleup and full automation of the culture system difficult. Other alternatives for plant propagation have been attempted using liquid culture. The liquid culture system offers several advantages of reducing labor and production costs by significantly lowering the subculture time using bigger containers and avoiding frequent culture vessel changes during media recharging. It also provides uniform controllable culture conditions with the possibility of scaleup and complete automation of the culture system [95]. 

Using liquid culture in the bioreactor system is one of the most cost-effective and efficient ways of in vitro plant propagation on a large scale by lowering production costs by more than 40% [97]. The liquid medium allows the plant tissues to have better contact for nutrient absorption enabling better culture growth and development [98]. However, the continuous submersion of plants under the liquid medium in a complete immersion bioreactor system produces unwanted hyperhydricity and asphyxia, undermining normal plant development [99]. Asphyxia is an unwanted physiological condition caused to plants due to low oxygen supply producing malformation in plants by affecting normal growth [100]. Hyperhydricity refers to the phenomenon of surplus accumulation of water in plants resulting in a glassy appearance due to the development of translucent, wrinkled, and brittle leaves with reduced chlorophyll content and abnormal chloroplast development [101,102]. Due to the aberrant leaf anatomy of hyperhydricity-affected plants, the success rate for their proper acclimatization in the greenhouse is usually low, leading to high losses in commercial plant propagation [95].
plants-12-01136-t001_Table 1Table 1Orchid micropropagation on semi-solid (SS) medium using different explants in the last four years.Name of the SpeciesDifferent Explants UsedCulture Conditions and Growth Hormone Combinations and Concentrations Generating Optimum Culture ResponseReferences*Anacamptis pyramidalis* (L.) Rich. and *Gymnadenia**conopsea* (L.) R. Br.SeedsMalmgren (MM) medium in complete darkness accelerated in vitro seed germination in both the orchids. MM + CW produced the highest germination (69–88%) in *A. pyramidalis,* but the maximum morphometric values of height (1.10 mm), width (1.00 mm), and bud height (0.57 mm) were obtained on MM + PE (peptone). MM + 0.3 mg L^−1^ 2iP gave the best plant development response in *G. conopsea* (plant height—5.33 mm; root number—1.13; root length—12.26 mm). [54]*Aerides multiflora*Roxb.Nodal and leaf segments from in vitro developed seedlingsShoot bud number from nodal explant was maximum (8.83 ± 0.45/segment) in MS + 1.0 mg L^−1^ NAA + 2.0 mg L^−1^ BAP. Longest shoot bud was obtained in MS + 1.0 mg L^−1^ NAA + 1.0 mg L^−1^ BAP. Root proliferation and development were superior in MS + 1.0 mg L^−1^ IBA. PLBs formation from the leaf explant was best in MS + 1.0 mg L^−1^ IAA + 2.0 mg L^−1^ BAP. The longest length (4.17 ± 0.13 cm) of individual shoot buds/PLBs after 30 days of culture was recorded in PM + 2% (*w*/*v*) sucrose + 0.5 mg L^−1^ NAA + 1.0 mg L^−1^ BAP.[103]*Brassavola nodosa* (L.)Lindl.In vitro shoot tips (0.3 to 0.5 cm)Out of the 6 hormonal treatments used for shoot multiplication, the third treatment (T3: 2.0 mg L^−1^ BA and 30.0 mg adenine sulphate) generated the highest shoot number per explant, while the survival rate witnessed for 6 treatments was almost similar. After transferring to the rooting medium, the plantlets showed maximum root formation on 0.5 mg L^−1^ NAA supplemented medium.[104]*Catasetum integerrimum* HookSeedsMS + 2.5 mg L^−1^ BAP + 5.0 mg L^−1^ IAA produced the highest shoots per explant (5.73 ± 0.45) and leaves per shoot (5.84 ± 0.48). MS + 2.5 mg L^−1^ IAA generated the best rooting response (11.20 ± 0.28 roots; 13.20 ± 0.28 cm root length). When plant parts (leaves, roots, and pseudobulbs) from in vitro seedlings were employed as explants, best leaf (5.50 ± 0.18) and root formation (4.37 ± 0.37) were achieved with pseudobulb explants.[105]*Cattleya gaskelliana*and *C. warscewiczii*SeedsSeeds after sterilization (chlorine 0.5%, chlorine 1%, distilled water, and sucrose) were checked for viability with 2 concentrations of tetrazolium (0.25% and 0.5%) and duration exposure of 24 h and 48 h. Seed viability was 90.6% for *C. gaskelliana* in 0.5% tetrazolium, while it was 90% for *C. warscewiczii* in 0.25% tetrazolium with exposure of 48 h in both treatments. The best seedling growth for *C. gaskelliana* and *C. warscewiczii* was witnessed in MS + CW and MS + P (pineapple juice), respectively, after 18 weeks of culture.[106]*Cattleya warneri* T. MooreSeeds Seeds germinated successfully on ½ MS + microalgal biomass or its aqueous extract (0.25, 0.5, 1.0, and 2.0 g L^−1^). Seed germination enhanced with the development of chlorophyllous protocorms at 4 weeks after the supplementation of biomass or extract (0.25 g L^−1^). The seedling development was high (greater than 95%) in all treatments with biomass and microalgal extracts (0.25 or 0.5 g L^−1^) after 24 weeks of culture. MS + 2.0 g L^−1^ AC produced elongated shoots and roots.[107]*Coelogyne ovalis* Lindl.Nodal budPLB formation was maximum (80%) in KC + 10 µM meta-Topolin + 0.5 µM NAA. Medium augmented with 10 µM IAA was the most suitable for rooting.[24]*Crepidium acuminatum* (D.Don) Szlach.Floral budsM + 1 mg L^−1^ IAA + 1 mg L^−1^ KN + 2% Sucrose + 2 g L^−1^ AC produced the highest shoot bud regeneration with 8 to 10 pseudobulbous shoots per floral bud.[108]*Cypripedium subtropicum*SeedsSeed germination accelerated on medium with 2iP or BA though the higher concentration of BA (4 and 8 μM) reduced seed germination. Medium with 2ip produced the highest surviving rate for protocorms compared with those with KN or BA. The highest seedlings developed after 4 months on Norstog medium fortified with 1 mg L^−1^ malic acid, 20 g L^−1^ sucrose, and 20 g L^−1^ potato homogenate and solidified with 7 g L^−1^ agar.[38]*Cymbidium eburneum* Lindley.Leaf segments of in vitro grown plants M + 0.5 mg L^−1^ BAP + 2.0 mg L^−1^ NAA produced a plant regeneration rate (83.3%) in 5.25 weeks generating 15.7 plantlets/explant after 30 WOC. M + 2 mg L^−1^ BAP + 2 mg L^−1^ NAA promoted PLB-mediated regeneration in 66.6% of the explants within 6.32 weeks.[109]*Dendrobium anosmum* Lindl.SeedsHigh protocorm formation (100%) was observed in all the concentrations of BAP or KN, alone or in combination with NAA, after 10 WOC. MS + 1.0 mg L^−1^ KN + 0.5 mg L^−1^ NAA + 30 g sucrose + 8.0 g L^−1^ agar was suitable for shoot length growth. The best rooting response (100%) was recorded in MS + 1 mg L^−1^ KN + 0.2, 0.3, or 0.5 mg L^−1^ NAA.[110]*Dendrobium chryseum* Rolfe.Seed derived protocorms½ MS + 2.0 mg L^−1^ KN + 10% CW produced highest shoot multiplication (18.75 ± 0.48 shoots/culture). MS + 1.0 mg L^−1^ GA_3_ + 10% CW yielded the longest shoots (2.0 ± 0.20 cm) and greatest shoot number (4.5 ± 0.65) per culture. Root growth and multiplication were best noticed on ½ MS +1.5 mg L^−1^ IAA.[111]*Dendrobium crepidatum* Lindley & Paxton SeedsThe highest protocorm formation (41 ± 0.76% for the late capsule; 36.33 ± 0.96% for the early capsule) was witnessed in ½ MS medium. Maximum plant growth and development were demonstrated when the germinated seeds were transferred to ½ MS + 2 µg mL^−1^ BAP + 1 µg mL^−1^ NAA.[112]*Dendrobium densiflorum* Lindl.Seeds½ MS +10% CW produced the highest seed germination. Root production from protocorms was maximum in MS +15% CW, while the greatest number of roots was noticed in MS +1.5 mg L^−1^ IBA. [113]*Dendrobium heterocarpum* Wall. ex Lindl.Immature embryos95% germination of embryos was observed on MS + sucrose (3%, *w*/*v*) + 3 mM L^−1^ kinetin. Shoot, root, pseudobulb length, and leaf and root number were found maximum on MS + 3 mM L^−1^ KN + 12 mM L^−1^ NAA + sucrose (3%). [49]*Dendrobium ovatum* (Willd.) KraenzlSeeds Seeds cultured on ½ MS +1 mg L^−1^ zeatin + 2% sucrose produced protocorms and PLBs after successful germination. ½ MS + 2 mg L^−1^ BAP was employed to grow the protocorms into plantlets. ½ MS +1 mg L^−1^ 2,4-D + 0.5 mg L^−1^ 6-BAP + 0.5 mg L^−1^ zeatin generated callus. Plantlets developed proper roots and shoots when transferred to ½ MS +1 mg L^−1^ zeatin + 2% sucrose.[114]*Dendrobium ovatum* (Willd.) KraenzlPLBs from in vitro germinated seeds MS + 1.0 mg L^−1^ TDZ + 0.5 mg L^−1^ NAA produced maximum induction of embryogenic callus (EC) (58.6%) and somatic embryos (SEs) (39.8/explant). The explants in the upright orientation gave a greater percentage of EC and higher SEs/explants (EC—58.6% and SEs—39.8/explants) compared to explants with inverted orientation, irrespective of growth hormone combinations. [115]*Dendrobium palpebrae* Rchb. fIn vitro derived pseudobulbs Through organogenesis, multiple shoot buds were developed from both the upper and lower parts of the pseudobulb. MS + 1.0 mg L^−1^ NAA + 2.0 mg L^−1^ BAP yielded the maximum shoot buds (8.21 ± 0.44) per segment in the lower part and the highest shoot buds (6.43 ± 0.40) per segment in the upper part of the pseudobulb. The longest root (4.82 ± 0.22 cm) and the greatest root number (2.75 ± 0.17) per shoot bud were recorded on MS + 0.5 mg L^−1^ NAA.[116]*Dendrobium Yuki* WhiteApical shoot segmentMS + 0.5 mg L^−1^ BA + 0.1 mg L^−1^ NAA + 40 mg L^−1^ adenine sulphate produced maximum shoots (12) and root number (17) per explant within 8 weeks of culture. The in vitro generated plants were acclimatized with 97% survival rate in charcoal blocks for 6 weeks, followed by plant transfer in potting mixture with coconut fiber and charcoal (1:1).[117]*Doritis pulcherrima* Lindl.In vitro derived protocormsNew Dogashima Media (NDM) was better than VW and MS media giving improved protocorm survival rate (46.70 ± 0.51%), number (11.00 ± 2.94 PLBs per protocorm), and size (6.35 ± 4.31 mm). NDM + 0.1 mg L^−1^ NAA + 0.1 mg L^−1^ BA produced maximum shoot, leaf, and root number and length. [118]*Dryadella zebrina* (Porsch) LuerSeeds The cultures with different BAP treatments showed a mean survival rate greater than 97%. However, BAP concentration higher than 9 μM significantly reduced plant survival. MS fortified with 6 μM BAP generated the highest shoot formation, while MS with 12 or 15 μM BAP yielded less shoots indicating its deleterious effect on shoot development at an elevated level. [119]*Encyclia cordigera* (Kunth) DresslerSeeds½ MS + AC (0.15%) produced the best germination response (100%), while ¼ MS + AC (0.15%) generated seedlings with the longest height (1.53 cm). MS + AC (0.15%) gave the maximum root number (2) and root length (2 cm). [50]*Epidendrum denticulatum* Barb. RodSeeds In vitro seed-derived plants were subjected to different LED types with blue/red (B/R) combinations for 90 days. White (W) light influenced the production of higher fresh and dry mass, while blue (B) light gave higher anthocyanins value under in vitro conditions. The total chlorophyll values were higher under B/R Light, and B and B/R wavelengths brought higher Fv/Fm values. [120]*Epidendrum fulgens* Brongn.Different explants (protocorm bases, leaf, and root tips) derived from in vitro seed-derived plantlets. The PLB induction was higher in MS +10 μM TDZ. PLB number increased with TDZ concentration higher than 15 μM, but induction frequency reduced at TDZ concentration greater than 10 μM. The most responsive explants for the highest PLB induction (90%) were protocorm bases with plants regenerated only in a single subculture to hormone-free medium in a shorter time (12 weeks) compared to other leaves (24 weeks) and root tips (60 weeks) explants.[33]*Eulophia dabia* (D. Don)Axenic rhizome segments½ MS showed 5% seed germination efficiency. MS medium with casein hydrolysate and AC produced the best rhizome growth from rhizome-like bodies. Maximum shoot induction (96.1%) response was witnessed on MS + 4.4 μM BAP +AC using axenic rhizome with maximum shoots (4.3) and shoot length (13.4 cm).[121]*Gastrochilus matsuran* (Makino)Seeds½ MS (without vitamins) + 1 µM NAA + 1.5 µM GA_3_ + 0.2% peptone + AC (0.05%) + 1% banana pulp + 3% sucrose + 0.8% agar gave the highest seed germination (93.3%). MS + 2 µM TDZ produced the best secondary protocorm formation. MS + 2 µM IBA or 1 µM NAA showed maximum conversion of protocorms into seedlings. [70]*Laelia anceps* ssp. *anceps*SeedsMS +2 mg L^−1^ BAP + 2 mg L^−1^ IAA + 2 mg L ^−1^ NAA gave the highest seed germination rate (82.20%), with seedlings exhibiting the highest leaf number (1.64) and length (1.11 cm) per explant. MS +1 mg L^−1^ IAA + 150 mL L^−1^ CW produced the best rooting percentage (78.20%). [122]*Ludisia discolor*NodalExplants treated with 0.40% HgCl_2_ produced the best survival (63.1%) and growth (22.5%) rate of the culture. ½ MS + 1.0 mg L^−1^ NAA + 0.1 mg L^−1^ TDZ + AC (0.2%) + 8% banana cultivar homogenate + 3% sucrose + 3.5 g L^−1^ generated maximum survival (42%) and plant growth rate (19.6%). [34]*Malaxis acuminata* D. DonTransverse thin cell layer segments (1–4 mm) excised from the pseudobulb Maximum shoot proliferation (21 micro-shoots/explant) was found in MS + 1.5 mg L^−1^ *meta*-Topolin (*m*T) + 5 mg L^−1^ chitosan. MS +1.5 mg L^−1^ IBA + 5 mg L^−1^ phloroglucinol (PG) produced the best rooting response (root number—7.22 ± 0.45; root length—3.62 ± 0.28 cm). [9]*Mokara* Sayan X *Ascocenda* Wangsa goldLeaf sectionsMS + 3 mg L^−1^ TDZ induced maximum PLBs (34 PLBs cm^−2^ leaf section), induction frequency (82.8%), and highest growth rate (93.7 mg day^−1^). The protocorms were best encapsulated at 3% sodium alginate and 75 mM calcium chloride. Furthermore, 71.2% germination frequency displayed by synthetic seeds stored at 25 °C even after 180 days, while those stored at 4 °C degenerated completely. [123]*Orchis militaris* L.SeedsmM (Malmgren modified terrestrial orchid medium) + CW (5%) + birch sap (5%) + AC (0.1%) produced the highest seed germination (82.6%). The seedling formation was through protocorm development without callus formation in all 3 modified culture media (Harvais, KC and Malmgren). Modified Harvais 2 medium was suitable for protocorm proliferation in darkness, while KC incorporated with AC was appropriate for further culture development leading to seedling formation. [48]*Orchis simia* LamSeedsmM (modified Malmgren medium) + pineapple juice (PJ) + casein hydrolysate (CH) gave the highest seed germination (94.51 ± 0.96%). mM + CW + AV yielded the quickest seed germination (6.8 ± 0.20 days), while mM + PJ with either AV (Aminoven) or CH made larger and higher-weight protocorms. Medium with PJ + AV generated the longest plantlet (4.2 ± 0.04 cm), shoot lengths (1.96 ± 0.042 cm), and heaviest weight (0.58 ± 0.002 g), while the maximum root formation was witnessed in medium with CW and AV (5.2 ± 0.20). [124]*Paphiopedilum SCBG Huihuang 90*(P. SCBG Prince × P. SCBG Miracle)Seeds Seeds germinated on Hyponex No. 26 medium + 0.5 g L^−1^ AC + 1.0 mg L^−1^ NAA. The protocorms produced meristem mass after transferring them to ½ MS + 0.05 mg L^−1^ 2,4-D. Higher level of IAA and jasmonic acid (JA) promoted PLBs differentiation, while lowering GA_3_ concentration was essential for shoot apical meristem (SAM) development.[37]*Paphiopedilum insigne*SeedsThe production of protocorm was high in ¼ MS +1 mg·dm^−3^ BAP + 2 mg·dm^−3^ TDZ (73%) and 5 mg·dm^−3^ KN + 1 mg·dm^−3^ BAP (67%). Both combinations revealed 99% leaf formation from protocorms. The fresh weight of regenerants was high (9.07 mg) in 5 mg·dm^−3^ KN +1 mg·dm^−3^ BAP.[36]*Paphiopedilum niveum* Rchb.f.SeedsThe highest percentage (68.33%) of somatic embryo formation was noticed in modified VC+ 0.1 mg L^−1^ NAA, with the production of the maximum number of somatic embryos (5.19 ± 0.67 per explant). High fresh weight accumulated (183.33 mg) in modified VW without NAA and KN.[69]*Phalaenopsis amabilis* (L.) BlumeShoot tips½ MS + 3% (*w*/*v*) sucrose + 0.1 g L^−1^ myoinositol, 2 mg L^−1^ thidiazuron + 1 mg L^−1^ BAP was employed to initiate culture with shoot-tip explants. Shoot multiplication was observed better under the blue + green light irradiation, but biomass accumulation was higher with white LEDs. The best shoot branching and multiplication were noticed with higher KN content, total cytokinins, and GA_3_ under blue + green lights. [82]*Phalaenopsis amboinensis* J. J. SmSeedsBest seed germination (90.7%) and protocorm development (51.4%) were witnessed on the VW medium. Leaf, root, and plantlet development was superior in medium augmented with 15% CW + 10 g L^−1^ banana homogenate (BH).[125]*Phalaenopsis pulcherrima* (Lindl.) J. J. SmLeaf segmentsVW + CW (2%) +100 g L^−1^ potato + sucrose (2%) +AC (0.2%) + 50 g L^−1^ banana extract + 3 mg L^−1^ thidiazuron, and ½ MS + 0.5 mg L^−1^ niacin + 0.1 mg L^−1^ thiamine–HCl + 0.5 mg L^−1^ pyridoxine–HCl + 100 mg L^−1^ myo-inositol + 2 mg L^−1^ glycine + banana extract (2%) + 3 mg L^−1^ thidiazuron was used for culture initiation using leaf explants. The maximum PLBs and highest PLB induction were observed under R (red): B (blue) LEDs on both MS and VW media. Shoot elongation, shoot number, and chlorophyll *a* and *b* content were promoted in response to R: B LEDs.[126]*Rhynchostylis retusa* (L.) Blume Root tipsRoot explants with intact tips and root caps with distal ends displayed good growth irrespective of the chemical regime. M + 3 mg L^−1^ KN + 1 mg L^−1^ NAA + 2% sucrose showed maximum culture regeneration (31%) in the proximal region of the root segment giving 28 plantlets in 15 weeks.[89]*Rhynchostylis retusa* (L.) Blume Immature capsules ½ MS and ¼ MS demonstrated the earliest seed germination and protocorm development. MS + 10% CW provided high shoot multiplication (12.8) and the longest shoot length (5.3 cm) in MS + 10% CW. The greatest root number (7.3) and root length (5.0 cm) were noticed in MS + fungal elicitor CVS4 extracted from the stem of *Vanda cristata*. [127]*Spathoglottis plicata* BlumeLeavesThe highest somatic embryogenesis (93.7%) was observed in MS +1.0 mg L^−1^ 2,4-D. Somatic embryo proliferation and shoot bud development were high in MS + 2.0 mg L^−1^ BAP. ½ MS + 1.0 mg L^−1^ IBA generated maximum rooting (93.6%) in ½ MS + 1.0 mg L^−1^ IBA. Synthetic seeds were best formed with somatic embryos encapsulated in 3% sodium alginate + 100 mM CaCl_2_.[128]*Spathoglottis plicata* BlumeSeeds High seed germination (93%) was observed on MS + 1.0 mg L^−1^ BAP, while lower germination was witnessed on MS with either KN, IAA, or NAA. The shoots obtained from liquid culture showed a better rooting response (94%), producing higher root numbers and lengths (13.0 ± 0.22 roots per shoot, 4.0 ± 0.25 cm length) compared to root development (6.5 ± 0.29 roots and 3.3 ± 0.19 cm length) observed on SS medium. [6]*Stanhopea tigrina* Bateman ex Lind.SeedsHigh seed germination (98%) was witnessed after 120 days of culture on MS + 1% AC. MS + 10 g L^−1^ apple extract or 10 g L^−1^ banana extract or 30 mL L^−1^ CW or 5.0 mg L^−1^ BAP were effective for shoot induction (1.25 ± 0.35). Maximum root formation (9.00 ± 0.68 roots) was achieved in MS + 5.0 mg L^−1^ IAA + 100 mL L^−1^ CW.[129]*Tolumnia* Snow FairyLeaf segments from in vitro plantsMaximum PLB formation was found in MS + 4.0 mg L^−1^ BAP + 0.5 mg L^−1^ NAA with an average of 24.0 PLBs. However, there was no PLB formation from the outer leaves; only the inner expanding leaves showed protocorm induction (25.5 PLBs per explant) on MS + 4.0 mg L^−1^ BAP + 0.5 mg L^−1^ NAA. The shoot generation rate from PLBs was 33.3% for the whole PLB, while upper PLB halves produced 40%. [75]*Vanda bicolor* GriffSeedsMS + 3 µM NAA + 3 µM BA showed 88.2% seed germination. The protocorms developed into plantlets with healthy leaves (6.2) and roots (3.3) on MS + 3 µM NAA + 6 µM BA + AC (0.6%).[19]*Vanda brunnea* Rchb.f.Shoot tipsA high plant regeneration rate (92–100%) was witnessed with Orchimax and MS medium supplemented with 0.5 mg L^−1^ BA. Orchimax showed the highest plant regeneration rate (100%), irrespective of the presence of BA in the medium. The number of plants obtained in Orchimax (6.2 per explant) was two times more than the plants produced (3.1 per explant) in MS.[130]*Vanda cristata* Wall. ex Lindl.Whole leafThe regeneration responses from the leaf explants were maximum (100%) on both M and KC medium fortified with NAA (10.6 µM) and BAP (8.8 µM). The highest number of plantlets (6) were obtained after explant differentiation via callus, PLBs, and shoot bud formation on the same hormonal combination. [131]*Vanilla planifolia* Jacks. ex AndrewsSeedsSeeds germinated successfully on ½ MS + 2 mg L^−1^ glycine + 0.5 mg L^−1^ niacin + 0.5 mg L^−1^ pyridoxine HCl + 0.1 mg L^−1^ thiamine + 1 g L^−1^ tryptone + 20 g L^−1^ sucrose + 7 g L^−1^ agar. The seed germination time increased from 75 to 90 min when the mature seeds were treated with 4% sodium chlorite solution before inoculation, and germination percentage was recorded highest with immature seeds collected 45 days after pollination. The seedlings developed after the protocorms were grown on ½ MS + 20 g L^−1^ sucrose + 1 g L^−1^ AC + 20 g L^−1^ potato homogenate + 7 g L^−1^ agar.[23]*Vanilla planifolia* Jacks. ex AndrewsLeaf segmentsThe leaf explants produced non-embryogenic calli on MS + 3.0 mg L^−1^ 2,4-D. The non-embryonic callus acquired embryogenic potential when transferred on MS + 1.0 mg L^−1^ BAP + 1.0 mg L^−1^ KN + 2.5 mg L^−1^ SN. Leaf explants, however, induced direct somatic embryogenesis (92.0%) on MS + 1.0 mg L^−1^ BAP + 1.0 mgL ^−1^ KN + 2.5 mg L^−1^ SN generating maximum somatic embryos (30.0 per explant). The embryos encapsulated and stored at −4 °C for 1 year demonstrated the highest germination (95.3 ± 0.49) and shoot multiplication (17.2 shoots per SE) on MS + 0.5 mg L^−1^ BAP + 0.25 mg L^−1^ KN + 2.5 mg L^−1^ SN. [25]*Vanda pumila* Hook.f.ProtocormsHighest shoots (9.50 ± 0.29) per culture formed on ½ MS + 1.0 mg L^−1^ KN + 10% CW. The shoot length was greatest (0.78 ± 0.07 cm) per culture on MS + 2.0 mg L^−1^ BAP + 10% CW. The ½ MS + 0.5 mg L^−1^ IAA produced high root formation (5 ± 0.00) per culture with good root length (0.93 ± 0.07 cm).[132]*Vanda tessellate* (Roxb.) Hook. ex G. DonSeedsMS gave the maximum seed germination (94%). MS + 2.0 mg L^−1^ BAP + 0.5 mg L^−1^ IAA produced the highest (89.4%) calli induction. Highest somatic embryo production (96%) from PLBs was observed in MS + 1.0 mg L^−1^ BAP + 0.5 mg L^−1^ IAA. Synthetic seed formation was best in MS + 2% sodium alginate + 100 mM CaCl_2_. Maximum germination (91%) of the cold stored encapsulated seeds was witnessed in MS + 50 mg L^−1^ ascorbic acid + 25 mg L^−1^ each of citric acid, adenine sulphate, and L-arginine + 0.5 mg L^−1^ each of BAP, KN, and IAA.[133]**Note:** ½ MS = half strength Murashige and Skoog medium; 2,4-D = 2,4-dichlorophenoxy acetic acid; CW = coconut water; BAP = 6-benzylamino purine; FT = foliar fertilizer; GA_3_ = gibberellic acid; IAA = indole-3-acetic acid; IBA = indole-3-butyric acid; KC = Knudson C medium; KN = kinetin; MS = Murashige and Skoog medium; M = Mitra medium; NAA = 1-naphthalene acetic acid; Pic = picloram; PLB(s) = protocorm-like-bodies; PM = Phytamax medium; TDZ = thidiazuron; MVW/VW = Vacin and Went; ZN = zeatin; SN = silver nitrate.

## 3. Orchid Micropropagation Using Temporary Immersion System (TIS)

TIS overcomes the limitations of SS and liquid culture systems by temporarily submerging the plants in the liquid medium for shorter periods, followed by exposing them directly to the gaseous environment by draining the liquid medium. The shorter immersion time and more prolonged gas exposure lower the detrimental effects of hyperhydricity and asphyxia on plants, providing optimal environmental conditions for efficient nutrient absorption under the least liquid contact [134]. Greater gas exposure improves oxygen transport to cultured cells minimizing oxygen limitation and lowering the asphyxia effect on the plant tissues in TIS [135]. Furthermore, enhancing headspace with carbon dioxide (CO_2_) and culture agitation by hydrodynamic forces without mechanical devices in TIS allows normal development and increases plant tissue multiplication with regular photosynthetic activities and minor shear stress [136]. Figure 2 gives a general overview of orchid propagation using TIS. Despite many benefits of TIS, the application of the SS system for orchid micropropagation cannot be overlooked as both the culture systems could be incorporated into the orchid propagation program for the effective mass production of these ornamental plants. Figure 3 shows the interrelationship of the two culture systems and how they can be employed for orchid micropropagation.

### 3.1. Important Factors Influencing Temporary Immersion System (TIS)

The culture medium provides nutrients required by the plants to grow and develop. Using a suitable medium in the TIS system is essential for successful plant propagation. Though MS medium is employed most frequently in TIS, the need for an appropriate culture medium depends on the nutrient requirement of plant species during its developmental period [137]. The volume of the culture medium is also crucial in influencing plant multiplication rate, leaf and root formation, and growth. The shoot multiplication rate was enhanced from 8.3 shoots (in 30 days) to 23.9 shoots (30 days) when the medium volume in the TIS bioreactor was increased from 5.0 to 50.0 mL per explant in *Saccharum* spp. [138]. Escalona et al. [139] also established the optimum medium volume for maximum shoot proliferation of pineapple at a higher volume of 200 mL per explant. Roels et al. [140] demonstrated the escalated shoot multiplication rate (11.9 to 13.8), shoot length (4 to 5 cm), leaf number (3.1 to 3.7), and root number (2.8 to 3.2) when the medium volume was raised from 10 to 30 mL in TIS. Uma et al. [141] observed the highest shoot production of bananas (24 shoots) in a 250 mL volume of the medium as compared to shoots generated (20 shoots) in other volumes (100 mL and 500 mL) tested. However, a larger culture volume than its optimum level was not favorable for plant growth as extracellular chemicals excreted from the cultures were diluted in a higher volume [137]. Container size of the TIS may also influence the plant growth as bigger vessels can accumulate larger medium volumes avoiding culture overcrowding and early shortage of nutrients. Monette [142] demonstrated longer shoot development in grapevine in bigger containers of square wide-mouth Mason jars (910 mL) compared to smaller Erlenmeyer flasks (125 mL). Krueger et al. [143] also revealed the advantage of larger culture vessels (7 L) over smaller baby food jars (140 mL) in imparting a positive influence on shoot elongation and multiplication by providing larger head space and lowering culture congestion.

TIS provides better culture aeration cultures than continuous and partial immersion liquid culture systems [135]. Exposure of the plants to gas is essential to avoid unwanted asphyxia in the culture [144]. Plants usually require different oxygen and carbon dioxide concentrations for proper development [99]. Bioreactors fitted with pressure and flow regulators can provide proper gas exposure to the cultures to enhance multiplication and growth. Increasing oxygen and carbon dioxide to a certain level is warranted as they are the crucial components of photosynthesis affecting plant growth and metabolism [99]. The oxygen transfer rate depends on its mass transfer coefficient, which is easily influenced by agitation and air flow rate and the design of the bioreactor [95]. The growth of seedless watermelons was enhanced when the oxygen level was raised by 40% inside the TIS [145]. The plants in the TIS bioreactor are temporarily submerged in the liquid medium for a short duration, during which they receive nutrients from the medium. When exposed to the gas after short immersion, the plants acquire oxygen and carbon dioxide, which are also essential for their development [95]. Optimizing the immersion time and frequency is vital as lengthening the immersion time will increase the hyperhydricity effect, and more prolonged gas exposure will enhance tissue drying and moisture loss, deterring normal development and proliferation [99]. The immersion time and frequency in TIS varied considerably with species and micropropagation process involved. Potato tuberization was effective when the plants were submerged for a longer duration (1 h) every 6 h, while somatic embryogenesis was accelerated with a brief immersion time of 1 min every 12 h in *C. arabica* [146,147]. The shoot multiplication rate of *Coffea* microcuttings also changed with different immersion times with plant submersion durations of 1, 5, and 15 min every 6 h producing multiplication rates of 3.5, 5.4, and 8.4, respectively [148]. Immersion frequency also influenced culture growth in TIS as immersion of 4 h six times per day produced maximum shoots (17.33) in *Dianthus caryophyllus* but with the highest rate of hyperhydricity [96]. The immersion of explants for 4 h four times daily produced the most desirable shoot growth (14.33 new shoots) without any hyperhydricity effect. Bello-Bello et al. [149] found the highest shoot multiplication rate (10.78 shoots per explant) of *Hylocereus undatus* in 2 min immersion time at 4 h intervals, while the least shoots (5.46 shoots per explant) were recorded in immersion frequency of every 16 h. Furthermore, the most extended shoots were formed at an immersion frequency of every 4 h, and the short shoots were noticed in other immersion frequencies (every 8, 12, and 16 h).

Explant inoculation density is one of the key factors that influence the growth of culture in the TIS. The optimum density of the explant should be determined as its increase in a culture vessel with constant medium volume may lead to poor aeration and congestion, affecting plant growth [149]. Pérez-Alonso et al. [150] showed potato microtuber formation improved from 168 to 234 when inoculum density from 60 explants per TIS was increased to 90 explants. However, there was an enhancement of the total fresh weight of microtubers per TIS (164.7 g) with less inoculum density of 60 explants compared to fresh weight (47 g) obtained with a higher density of 90 explants [150]. García-Ramírez et al. [151] also demonstrated the effect of inoculum density on the physiological development and morphology of *Bambusa vulgaris* shoots by taking 6, 12, and 18 explants per TIS. The inoculum density of 12 explants was more favorable for shoot growth as shoot and leaf number, shoot length, and chlorophyll content increased. Nevertheless, higher inoculum density was not suitable for culture growth with the accumulation of less total chlorophyll, lower dry mass, and water content of the shoots. Posada-Pérez et al. [152] employed inoculation densities of 100, 200, and 300 somatic embryos per TIS of papaya to determine the optimum density for best plant growth. The maximum response for callus, leaf and root formation, and root length were witnessed with the inoculation of 100 somatic embryos per TIS. Uma et al. [141] used varied inoculum densities (3, 6, and 12 explants) per TIS to ascertain the optimum inoculum density for the best shooting response in bananas. The TIS with explant densities of 3 and 6 generated higher shoots (24 shoots/explants) than those with inoculum densities of 12 explants.

### 3.2. Benefits and Drawbacks of TIS

The benefits of using TIS over a conventional SS system are enormous in terms of lowering the production and labor cost by preventing frequent subculturing using higher culture liquid volume and containers, enhancing plant multiplication rate by accelerating nutrient absorption through uniform intermittent contact with liquid medium and plant tissues, normal morphological plant development by reducing hyperhydricity and asphyxia effects, and the possibility of successful large scaleup with full automation of culture systems [95,141,145]. Lorenzo et al. [138] reported 46% reduction in the production cost of shoot multiplication of *Saccharum* spp. using TIS compared to the conventional SS culture system. Pineapple generation using TIS produced a 100-fold increase in shoot generation and lowered the production cost by 20% compared with standard liquid medium. Bello-Bello et al. [149] established in vitro protocols for scaling-up *Pitahaya* propagation using TIS, which produced a multiplication twice that of semi-solid and partial immersion media. Ducos et al. [153] also succeeded in scaling up culture production by generating 2.5 million pregerminated embryos of *Coffea canephora* annually in a 40 m^2^ size culture room using 100 TIS units of 10 L volume each. Ptak et al. [154] demonstrated the positive influence of metatopolin and benzyladenine on plant development with two times more production of morphologically normal plants from SEs in TIS than SS medium without any hyperhydricity. TIS also helps reduce medium browning due to oxidation, plant contamination through air vents, and lower agitation stress on plant tissues due to the absence of mechanical devices [155,156]. Though TIS is mainly utilized for plant micropropagation, it also offers an alternative to SS and liquid culture for increasing biomass yield and secondary metabolite production at a low cost. Pavlov and Bley [157] found high biomass accumulation (18.8 mg g^−1^) and betalains yield (9.6 mg g^−1^) when hairy roots of *Beta vulgaris* were cultured in TIS at immersion frequency of 15 min immersion every 60 min interval. Kokotkiewicz et al. [158] reported the TIS to be more effective in phenolic secondary metabolite accumulation (xanthones and benzophenone derivatives) in the cell culture of *Cyclopia genistoides* than liquid culture systems providing a possible cheap alternative source of the phytochemicals from plants. Kunakhonnuruk et al. [159] found the production of biomass per clump of *Drosera communis* (3.40 g fresh weight and 0.36 g dry weight) in TIS thrice and 1.8 times greater than SS and continuous immersion systems, respectively. Furthermore, the maximum plumbagin yield per replication (17.31 μg/replication) was attained in cultures grown in TIS compared to those in SS and continuous culture systems. TIS has also been applied for in vitro production of foreign proteins in plant cells and tissues. Michoux et al. [160] obtained high expression of a modified form of the green fluorescent protein (GFP^+^) and a vaccine antigen, fragment C of tetanus toxin (TetC), in transgenic cells cultured in TIS with thidiazuron for inducing proper shoot initiation. The yield of GFP^+^ (660 mg L^−1^ of bioreactor) and TetC (95 mg L^−1^) were much higher than protein expression witnessed in transformed cells in suspension cultures.

Despite having the many advantages of TIS, it also includes one of the main limitations of difficulty in scaling up to the commercial scale. The volume of containers involved in the TIS can be increased to 10–20 L, which is still less for plant propagation at the commercial level. However, the utilization of larger vessels might hamper the performance of TIS as the use of 10 L jars did not produce normal embryo development of *C. canephora* due to uneven light distribution [153]. The increased plant biomass in bigger culture containers disrupted light penetration and resisted nutrient and oxygen transfer [153,161]. Utilization of larger culture vessels may not necessarily be an effective approach to scale up the process for plant propagation. One way of overcoming this shortcoming is by using several smaller-size containers (1–5 L) that may be subjected to simultaneous operation under preset culture parameters in an automated fashion that will ensure plant propagation at a commercial scale. The TIS bioreactors should be designed to provide a particular microenvironment conducive to the growth and development of complex differentiated plant tissue and organs [162]. Many TIS with varying designs are developed to meet the specific culture requirements for large-scale plant multiplication. Some popular TIS available in the markets are Twin-Flask, Ebb-and-Flow, RITA, Thermo-photobioreactor, BioMINT, SETIS, and PLANTIMA [163]. However, these commercially available TIS are associated with several drawbacks, such as complex automation, unsuitable for forced ventilation and CO_2_ enrichment, low headspace, humidity in the growth chamber, no nutrient medium renewal, occupation of more space in the growth chamber, tilting platforms requirements, difficulty in biomass harvesting, complex automation and construction, and high cost and energy requirement [163]. Attempts have been made to refine and improve the TIS bioreactors circulating in the markets so that the existing shortcomings are eliminated and more effective, cheap, simple, and easy to store and handle compact bioreactor designs with autoclavable and reusable plastic elements with options for multiple uses are readily available.

### 3.3. Application of TIS for In Vitro Orchid Propagation 

The first instance of the application of TIS for orchid culture was evident when Tisserat and Vandercook [164] established an automated plant culture system (APCS), which did away with the necessity of the frequent manual subculturing of the culture to freshly prepared medium after every 4 to 8 weeks in either agar solidified or liquid medium. The shoot tips of *Potinera* orchid hybrid were subjected to different immersion frequencies to determine the optimum immersion frequency for the best plant growth response. Orchid tips subjected to less immersion frequency (4 and 1 immersion cycles/day) showed a lower survival rate when compared to those with a higher immersion frequency of 12 or 24 immersion cycles/day. When grown in an automated culture system, the orchid tips generated higher tissue mass than those in the agar-gelled medium. The orchid tissue production employing the APCS was four times higher in nine months than in the tissue mass generated in the same period on agar solidified medium.

Young et al. [165] used TIS and a continuous immersion system (CIS) to multiply PLBs of *Phalaenopsis* orchid from PLB explants obtained from in vitro grown leaf segments. TIS with the charcoal filter attached produced maximum PLBs (about 18,000) in eight weeks of incubation from 20 g of inoculum on 2 L Hyponex medium. The different aeration rates at 0.5 or 2.0 volumes of air per volume of medium min^−1^ (vvm) did not impact the PLB multiplication as they generated a similar amount of biomass. Liu et al. [30] used air-driven periodic immersion (API) bioreactors to culture PLBs obtained from lateral buds of the flower stalks of *Doritaenopsis.* Comparison of PLB growth on solid, liquid, and API showed the highest PLB proliferation and growth in the API system. PLBs growth increased by 4–6 fold when immersion time was set for 5 min at 4 h intervals. The increased level of PLBs formation in the API system may be due to the combined nature of both solid and liquid cultures. Yang et al. [136] also examined the feasibility of producing PLBs in bioreactors using shoot tips derived from in vitro plantlets of *Oncidium* ‘Sweet Sugar’. To initiate bioreactor cultures, 30g fresh weight (FW) PLB pieces were used, with PLBs being submerged in the medium throughout the culture period in the CIS while an immersion period of 1 h after every hour was applied for the TIS. The two bioreactor systems exhibited different growth rates of PLBs with a maximum growth ratio (10.9) witnessed in the CIS. Superior PLB proliferation may be due to the increased rate of nutrient uptake by the cultures because of their constant contact with the medium in the continuous immersion system. The earlier studies showed TIS appropriate for shoot multiplication [166,167], while the CIS is suitable for culture of storing organs such as adventitious roots [168,169], bulblets [170], and microtubes [171]. Ekmekçigil et al. [172] applied a thin cell layer and RITA temporary immersion bioreactor to mass propagate PLBs and shoots of *Cattleya forbesii* at the commercial level. The highest PLB production (PLBs per RITA—2237 PLBs and per explant—111.9 PLBs) was recorded when 20 tTCL-PLB explants were inoculated in 250 mL of medium with an immersion frequency of 1 min/4 h. Similar inoculum density in lower medium volume (150 mL) at the same immersion frequency generated the maximum shoot formation (shoots per RITA—3998 shoots and shoots per explant—199.9 shoots). Fritsche et al. [173] also found TIS favorable for PLB multiplication in *Cattleya tigrina* with MS medium incorporated with 30 g L^−1^ sucrose and Morel vitamins. Fresh weight increment rate (FWI) of PLB formation was significantly improved in TIS with a 2-fold increase in PLB proliferation (77.3 g) in comparison to that of PLB formation (35.3 g) in a continuous immersion system on the gelled medium.

Hempfling and Preil [174] used in vitro grown shoots of *Phalaenopsis* cv. Jaunina as inoculum for adventitious shoot multiplication and rooting in TIS. The shoot formation was maximum at 25.4 after 12 weeks of culture with an immersion frequency of eight immersion per day and immersion time of 10 min each. An increase in the time interval of medium recharge to four weeks significantly reduced the shoot multiplication rate to 14.5. The rooting response was tested by exposing shoots of 4–7 cm from TIS cultures to TDZ-free medium incorporated with 0.5 and 1.0 mg L^−1^ IAA or NAA. Maximum shoots were rooted (93.8%) with the production of the highest root number (3.7 roots per shoot) in medium supplemented with 1.0 mg L^−1^ IAA after subjecting to six immersions per day with an immersion time of 10 min each. Pisowotzki et al. [175] investigated the effect of PVPP on the in vitro shoot development of *Phalaenopsis*-hybrids grown in TIS. They observed that the phenolic compounds extracted from liquid and plant tissues exhibited the same peak pattern after HPLC separation, though the concentration of these compounds varied between different tissues, with their concentration higher in shoots than leaves. There was a reduction in biomass production and shoot proliferation rate when the phenolic compound was withdrawn from the culture medium, indicating its positive influence on culture growth. Biomass generation and shoot multiplication were higher in TIS system than in the conventional SS. Ramos-Castellá et al. [176] investigated the efficiency of TIS (RITA) for shoot multiplication of *Vanilla planifolia*. The highest shoot multiplication rate (14.27 shoots per explant) was best observed in TIS, applying an immersion frequency of 2 min every 4 h. The most appropriate medium volume for shoot multiplication was determined at 25 mL per explant, delivering the highest multiplication rate of 17.54 ± 1.14 shoots per explant. The TIS and partial immersion system did not produce a significant change in shoot length (12.67 ± 1.2), but solid medium generated the shortest shoots (10.47 ± 1.01 mm). The shoots were successfully rooted in TIS when transferred to ½ MS medium enriched with 0.44 NAA at an immersion frequency of 2 min every 4 h. Ramírez-Mosqueda and Iglesias-Andreu [29] employed three different TIS, viz., Recipient for Automated Temporary Immersion (RITA), Temporary Immersion Bioreactors (BIT), and Gravity Immersion Bioreactors (BIG) for establishing efficient in vitro protocols for *Vanilla planifolia* micropropagation. Shoot multiplication was the highest (18.06 shoots/explant) in BIT systems compared to shoot formation observed with RITA (12.77 shoots/explant) and BIG (6.83 shoots/explant). However, the maximum shoot length was witnessed in BIG (1.69 cm) and RITA (1.64 cm) compared to the BIT system. The shoots were rooted effectively in the three TIS with the highest number of roots noticed in BIT (4.27 roots/explant), followed by BIG (2.76 roots/explant) and RITA (1.90 roots/explant). Ramirez-Mosqueda and Bello-Bello [177] established commercially applicable in vitro propagation protocols for *Vanilla planifolia* by employing SETIS bioreactor. This bioreactor differs from RITA, BIT, and BIG systems in having horizontally placed larger-capacity vessels (4000 mL) with more headspace for plant development, enabling easy scaleup to commercial scale. Plant height and leaf number were recorded highest in TIS at 4.24 cm and 3.72 per explant, respectively. TIS also reported elevated percentages of closed stomata and a stomatal index indicating a high functionality of stomata, favoring a low transpiration rate. The survival rate of the in vitro propagated plants after acclimatization was also high (98%) for TIS.

Zhang et al. [32] examined culturing factors involved in pseudobulb development and plant growth by growing 6 mm long seedlings obtained from in vitro germinated seeds of *Bletilla striata* in a TIS. The biggest stem diameter was observed in immersion frequency of 3 min immersion every 6 h while the largest pseudobulb (4.88 ± 0.17 mm) was noticed in 3 min per 2 h interval. However, changing the immersion frequency from 2 to 4 h gave longer plant height but no significant increase in pseudobulb diameter. The optimum inoculum density was also determined at 300 explants per treatment, with plantlets showing the largest stem diameter (2.31 ± 0.08 mm) and the greatest length (141.27 ± 6.48 mm). Leyva-Ovalle et al. [178] developed an efficient in vitro protocol for *Guarianthe skinner* using TIS. The highest shoots per explant (6.06 ± 0.17) were noticed in TIS compared to SS medium (2.96 ± 0.18.) and partial immersion (1.70 ± 0.19). The 4 h interval was most favorable for plant development when different immersion frequencies (4, 6, and 8 h, with 2 min per immersion) were tested in the TIS system. The number of subculture cycles (subculture performed after every 30 days) impacted shoot number formation, with the maximum shoot number (16.56 ± 0.86) generated in the third subculture. The longest shoots (3.67 ± 0.19 cm) and leaf number (3.76 ± 0.10 leaves) were recorded in a rooting medium of ½ MS medium with GA_3_. Kunakhonnuruk et al. [179] established efficient in vitro protocols using TIS for large-scale propagation of *Epipactis flava*. The plants generated in TIS with an immersion frequency of 5 min every 4 h interval were healthier (53.3%) than SS (20.0%) and the continuous immersion system (CIS) (5.5%). The percentage of the new shoot (96.7%) and bud formation (91.7%) in TIS were higher compared to those in SS (46.7%) and CIS (40.0%). The plantlets derived from the TIS showed the highest survival percentage (76.7%) compared to SS (28.9%) and CIS (23.3%) after proper acclimatization. 

Yoon et al. [180] studied factors involved in biomass production using 1.0 to 1.5 cm long in vitro grown shoots of *Anoectochilus formosanus* as explants in bioreactor systems. The most appropriate inoculum size for best shoot proliferation was 8 g L^−1^, while maximum biomass accumulation was witnessed in the Hyponex medium augmented with 3% sucrose, coconut water (50 mL L^−1^), and AC (0.5 mg L^−1^) under a PPFD of 50 μmol m^−2^ s^−1^ light condition. The TIS was programmed with an immersion duration of 30 min four times daily. Zhang et al. [181] determined immersion frequency (min/h; 5/6) to be most suitable for biomass production and the highest total alkaloid content for in vitro *Dendrobium nobile* seedlings developed on 10 μM MeJA treated medium in TIS. TIS with MeJA generated maximum alkaloid content (7.41 mg g^−1^ DW) and production (361.24 mg L^−1^) when compared to those of TIBS without Meja (3.20 mg g^−1^ DW, 174.34 mg L^−1^). The in vitro propagation of different orchids using TIS is briefly described in Table 2.

## 4. Conclusions

Orchid micropropagation using the SS system is one of the promising methods of plant propagation on a large scale. However, expensive production costs, low multiplication rates, and detrimental effects of hyperhydricity and asphyxia limit the use of the system for mass plant generation. The use of TIS for orchid propagation reduces the production cost, does away with physiological problems of hyperhydricity and asphyxia, and makes scale-up and complete automation of the system highly feasible. However, the scale-up of TIS may require large, air-conditioned spaces, which may significantly increase production costs. The efficiency of both SS and TIS for orchid micropropagation cannot be questioned, but the refinement and improvement of both systems are the need of the hour for their commercial application in plant production. With more efforts being put into developing more effective, cheap, and reliable TIS, the objectives of mass orchid production for commercial and conservational purposes may soon be realized. 

## Figures and Tables

**Figure 1 plants-12-01136-f001:**
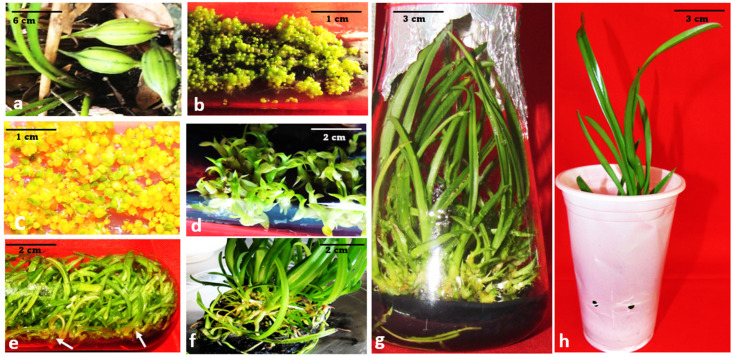
Micropropagation of *Cymbidium bicolor* on semi-solid (SS) medium using seeds. (**a**) Capsule as seed explant source. (**b**) Germinated seeds developing into spherical protocorms. (**c**) Shoot initiation from protocorms. (**d**) In vitro shoot growth and proliferation. (**e**,**f**) Root development and multiplication. (**g**) Well grown plantlets appropriate for hardening. (**h**) Acclimatization and hardening of in vitro propagated orchids.

**Figure 2 plants-12-01136-f002:**
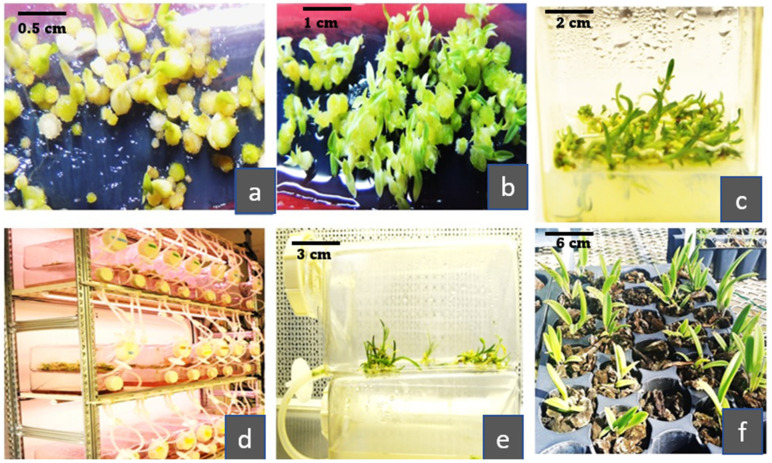
(**a**) In vitro orchid seed germination. (**b**) Protocorm formation. (**c**) Shoot multiplication to derive inoculum for temporary immersion system (TIS) culture initiation. (**d**) SETIS temporary immersion bioreactor. (**e**) Plant multiplication in TIS producing multiple shoots and roots. (**f**) Acclimatization of orchids propagated through TIS.

**Figure 3 plants-12-01136-f003:**
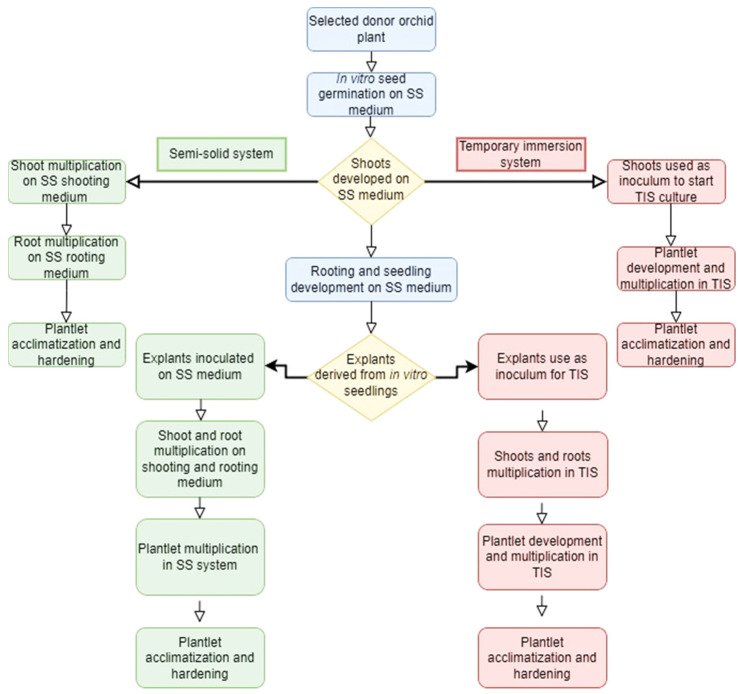
The application of SS and TIS for orchid micropropagation.

**Table 2 plants-12-01136-t002:** Micropropagation of different orchids using temporary immersion system (TIS).

Orchid Species	Explant Type	Culture Systems	Type of Medium and Hormone Combinations Used	Medium Volume	Immersion Time and Frequency	Experimental Results	References
*Cattleya tigrina* A. Rich. ex Beer	Homogeneous shoots (≥5 mm) with leaves and adventitious roots	CIS (continuous immersion system) and TIS	Liquid MS fortified with 30 g L^−1^ sucrose and Morel vitamins (2 g L^−1^ Phytagel added for CIS)	30 mL for CIS; 200 mL for TIS	-	The inoculated shoots gave rise to PLB directly and continued to proliferate without growth hormones. PLB multiplication significantly enhanced in TIS with 2-fold higher production (77.3 g) of PLBs) than those formed on the gelled medium of CIS (35.4 g PLBs).	[173]
*Cattleya walkeriana* Gardner	In vitro seedlings (1 cm long)	SSS (semi-solid system), liquid, CIS), and TIS	Liquid MS augmented with 1.34 μM NAA, 30 g L^−1^ of sucrose (6 g L^−1^ of agar added for solid medium).	60 mL for solid and liquid medium; 600 mL for CIS and TIS.	An immersion period of 3 min every 90 min.	The longest aerial part (2.06 cm) and biggest fresh mass (0.032 g) of the propagated plant were observed in the TIS. Furthermore, the largest fresh mass was noticed in the CIS and TIS bioreactors due to the continual contact of the explants with the medium. The TIS performed best as compared to other culture systems.	[182]
*Cymbidium sinense* Willd	Rhizome segment	CIS and TIS	Medium containing g L^−1^ Hyponex I, 0.5 g L^−1^ Hyponex II, 1 g L^−1^ peptone, 2 mg L^−1^ BA, 0.2 m L^−1^ NAA, 0.2 mg L^−1^ AC and 30 g L^−1^ sucrose for CIS; medium with 2 g L^−1^ Hyponex I, 0.5 g L^−1^ Hyponex II, 1 g L^−1^ peptone, 4 mg L^−1^ BA, 0.2 mg L^−1^ NAA and 30 g L^−1^ sucrose for TIS.	2000 mL for CIS and TIS.	1 h immersion with a drying period of 1 h.	Shoot induction from rhizome failed in CIS, unlike TIS, which produced the best shooting response and plantlet generation in medium appended with 4 mg L^−1^ BAP and 0.2 mg L^−1^ NAA. The root formation rate was prominent (94.7%) in the medium enriched with 1.0 mg L^−1^ NAA.	[31]
*Dendrobium candidum* Wall ex Lindl.	PLBs from the nodal stem segment	Raft type (protocorms cultured on the net), CIS and TIS (ebb and flood)	½ MS incorporated with 0.5 mg L^−1^ NAA, 2.5% (*w*/*v*) sucrose, 150 mg L^−1^ NaH_2_PO_4_ and 1% (*v*/*v*) banana homogenate.	2000 mL		The fresh and dry biomass accumulation was highest (323.33 g L^−1^ and 16.13 g L^−1^) in CIS while it was least (270.60 g L^−1^ and 14.67 g L^−1^) with ebb and flood method demonstrating better protocorm growth in immersion cultures. Accumulation of bioactive compounds was maximum (polysaccharides—404.48 mg g^−1^ DW, coumarins—18.36 mg g^−1^ DW, polyphenolics—13.33 mg g^−1^ DW, and flavonoids—3.97 mg g^−1^ DW) in immersion cultures. An inoculum density of 50 g L^−1^ was appropriate for biomass and bioactive compound accumulation in cultures.	[183]
*Dendrobium* *nobile* Lindl.	Protocorms from in vitro germinate seeds	SSS and TIS	The liquid ½ MS medium containing 0.5 mg L^−1^ NAA, 2% sucrose and 10% CW (0.7% agar added for SS culture)	1000 mL	5 min immersion for every 2, 4, 6, and 8 h duration.	The total fresh weight was the highest (302.85 g) with 6 h immersion frequency, while dry matter content (11.56%) was the maximum with 8 h immersion frequency. The longest shoot (72.83 mm), highest internode number (4.52), and largest stem diameter (4.05 mm) were achieved in 6 h of immersion frequency in the TIB. After acclimatization in the greenhouse, the shoot growth and plant survival rate were better with plants obtained from TIS than with the plants generated through SSS.	[184]
*Epidendrum* *fulgens* Brongn.	In vitro generated plantlets	Natural ventilation system (NV) and TIB system	Liquid MS appended with 3% sucrose for TIS (gelled media with 2 g L^−1^ phytagel poured into polypropylene containers containing PTFE filters allowing NV at a rate of 54 dm^3^ day^−1^ for NV culture system).	400 mL of gelled media for NV system; 400 mL liquid media for TIB system	3 min immersion every 3 h duration	The two culture systems (NV and TIBS) significantly affected plant growth and quality. The number of leaves, shoots, roots, and fresh weight was greater for plants developed in TIBS than those generated in the NV system. There was a 2-fold increase in root number for plants grown in TIBS compared to those in NV system, even though there was a significant impact on stomata number and photosynthetic pigment contents	[33]
*Mokara Leuen* Berger Gold	Callus tissue	SSS and TIS	MS incorporated with BA (0.5 mg L^−1^), B1 (5 mg L^−1^), adenin sulfate (10 mg L^−1^), peptone (1 g L^−1^), CW (10%) and sucrose (20 g L^−1^) for SSS; MS supplemented with CW (10%), sucrose (20 g L^−1^) and other growth regulators at different concentrations for TIS.	10 m for SS and 250 mL for TIS	1 min immersion time for 1 h interval	The callus proliferation was more prominent in TIS than SSS on MS supplemented with CW (30%), sucrose (30 g L^−1^), and 2.4 D (1 mg L^−1^). The shoots were rooted best on MS augmented with NAA (1 mg L^−1^), B1 (5 mg L^−1^), CW (10%), and sucrose (20 g L^−1^).	[185]
*Paphiopedilum rothschildianum* Rchb.f.	Callus induced from seeds and protocorm explants	SSS and TIS	½ MS enriched with 0–22.6 µM 2,4-D and 4.54 µM TDZ for SS culture; ½ MS appended with 2.27 µM TDZ and 12.0 µM BAP for TIS.	150 mL	Immersion time of 5 min after every 125 min	Callus proliferation in TIS produced a 3-fold increase in fresh weight compared to that cultured on SSS. Protocorm development from callus explant increased 3-fold in TIS with a regeneration capacity of 168 PLBs per gram calli. PLB regeneration capacity was enhanced further with increased sucrose concentration (15 µ to 58 µM) with the generation of 190 PLBs per gram calli.	[186]
*Paphiopedilum rothschildianum* Rchb.f.	Callus derived PLBs	SSS and TIS	MS supplemented with 4.54 μM TDZ singly or in association with 13.6 μM 2,4-D.	150 mL	5 min immersion after every 125 min.	Higher sucrose concentration promoted better PLB formation (4.0) on TIS, contrasted with greater PLBs formation in lower sucrose concentration in SSS. A 2-fold increase in PLB formation was observed on TIS compared to SSS, producing 168 PLBs per gram calli. Furthermore, the regeneration capacity in TIS enhanced to 190 PLBs per gram calli with higher sucrose concentration (58 mM).	[187]
*Phalaenopsis*	Shoot apical meristem	TIS	The liquid medium with 1.5 g L^−1^ Hyponex I, 0.1 mg L^−1^ NAA, 4 mg L^−1^ BAP, 200 mL L^−1^ CW and 20 g L^−1^ sucrose.	250 mL	10 min immersion every 4 h each time	The production of axillary shoots correlated with days in the culture. After 5 months of culture, a single virus-free short shoot segment generated around 200 plantlets on average. Shoots were separated into single and transferred to root initiation medium to get complete plantlets in 3–4 months.	[188]
*Vanilla planifolia* Jacks	In vitro shoots	Automated TIS (RITA)	MS supplemented with 2 mg L^−1^ BA with different concentrations of Argovit (0, 25, 50, 100 and 200 mg L^−1^).	200 mL	2 min per immersion with a time interval of every 4, 8, and 12 h.	Maximum shoots (14.89) per explant were recorded at Argovit doses of 50 mg L^−1^, while the least shoots (4.55) were observed at Argovit doses of 200 mg L^−1^. The highest shoot length (14.89 cm) was noticed at 50 mg L^−1^ of Argovit, and the shortest (0.82 ± 0.6 cm) was witnessed in Argovit doses of 200 mg L^−1^. Fresh weight was maximum (438.00 ± 18.42 mg) in shoots under 50 mg L^−1^ dose of Argovit and the lowest (143.80 ± 12.34 mg) was noted in 200 mg L^−1^ of Argovit.	[189]
*Vanilla planifolia* Jacks	Shoot nodal segments	TIS	MS basal medium supplemented with 30 g L^−1^ sucrose and 2.15 mg L^−1^ BA.	25 mL of medium per explant.	2 min immersion every 8 h.	The longest (2.79 cm) and maximum shoot number (9.15) per explant were recorded after the first subculture, and the least (4.57) shoots were achieved in the fourth subculture. 100% of the roots were rooted with 80% survival, but all showed variegation. 100% genetic uniformity was observed from molecular analysis with ISSR markers and morphological stability was evident from the heritability of leaf pigmentation.	[190]
*Vanda tricolor* Lindl.	In vitro grown shoots	Thin layer and TIS	MS basal medium fortified with 150 mL L^−1^ CW, and 30 g L^−1^ sucrose.	250 mL for TIS and 5 mL for thin layer system	Immersion time of 5 min and 10 min every 12 h interval for TIS; incubated in continuous 120 rpm in a gyratory shaker for thin layer system.	The sugar utilization in the thin layer system was more (25.28%) than in the TIS bioreactor (6.31%), resulting in a higher growth rate and biomass accumulation. The largest biomass production was noticed in the thin layer system, with a growth rate of 0.056 cm per day. However, the ability to sustain shoot viability and survivability was higher in TIS compared to the thin layer system.	[191]

## Data Availability

Not applicable.

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
