# Peer review of "Orchid Micropropagation Using Conventional Semi-Solid and Temporary Immersion Systems: A Review"

_plants, 2023, doi:10.3390/plants12051136_

Round 1

Reviewer 1 Report

Dear authors,

I have reviewed your manuscript "Orchid Micropropagation Using Conventional Semi-solid and Temporary Immersion Systems: A review" submitted for publication in Plants. The review manuscript provided updated information on the use of conventional semi-solid and temporary immersion systems for in vitro propagation of orchids. Overall, the manuscript is well written and informative. Some sentences require the addition of references as recommended below.

I have highlighted some minor points for your consideration and manuscript improvement:

Please double-check the units. "mgL" should be "mg L".. the same for other units.

It is necessary to add spacing between numbers and units; check this throughout the manuscript.

“®” should be in superscript form.

Line 14: Add author's abbreviation name

Line 50: …. labor-intensive (add a reference to support this sentence)

Line 79: …. pathogen-free (add a reference to support this sentence)

Line 102: at high auxin concentrations (add the concentration)

Line 104: same as above..

Line 108: different orchids (add references to support this sentence)

Line 115: (add a reference to support this sentence)

Line 139: “M medium” or “MS medium”. Please check it through the manuscript

Line 175: Describe TIS in the subtitle

Line 177: orchids (add references to support this sentence)

Line 195: Add scale to figures. It is necessary to improve the pic quality in "c", "d", and "f". Describe abbreviations in the figure caption

Line 223: Describe “CO2”

Line 259: Add scale to figures. Describe abbreviations in the figure caption

Line 265: Describe abbreviations in the figure caption

Line 267: Describe TIS in the subtitle. Please check it through the manuscript

Line 284: a higher volume (add a reference)

Line 293: culture systems (add references to support this sentence)

Lines 295, 300, 305, 318, and 331: again…. It is necessary to add a reference to support this sentence

Line 327: microtuber

Lines 347-353: Long sentence and requires adding references.

Line 391: transfer [153,161].

Line 400-401: Are these commercial brands? Is there any reference?

Lines 406-410: Long sentence and requires adding references.

Line 567: Describe all the table abbreviations in the table footer. 

Author Response

Please double-check the units. "mgL" should be "mg L".. the same for other units. Units double-checked and corrected.

It is necessary to add spacing between numbers and units; check this throughout the manuscript. Checked and corrected.

“®” should be in superscript form. Corrected.

Line 14: Add author's abbreviation name. Name abbreviation added

Line 50: …. labor-intensive (add a reference to support this sentence) Reference added.

Line 79: …. pathogen-free (add a reference to support this sentence)Reference added.

Line 102: at high auxin concentrations (add the concentration) Concentrations added

Line 104: same as above.. Concentrations added

Line 108: different orchids (add references to support this sentence References added.

Line 115: (add a reference to support this sentence) Reference added.

Line 139: “M medium” or “MS medium”. Please check it through the manuscript M medium defined (Mitra)

Line 175: Describe TIS in the subtitle TIS described in subtitle

Line 177: orchids (add references to support this sentence) References added.

Line 195: Add scale to figures. It is necessary to improve the pic quality in "c", "d", and "f". Describe abbreviations in the figure caption Scale bars added. Unfortunately, we do not have better resolution for photos c, d and f.

Line 223: Describe “CO2” Described in the text

Line 259: Add scale to figures. Describe abbreviations in the figure caption Scale bars added

Line 265: Describe abbreviations in the figure caption Abbreviations described

Line 267: Describe TIS in the subtitle. Please check it through the manuscript TIS described in subtitle

Line 284: a higher volume (add a reference)Reference added.

Line 293: culture systems (add references to support this sentence References added.

Lines 295, 300, 305, 318, and 331: again…. It is necessary to add a reference to support this sentence References added.

Line 327: microtuber Corrected

Lines 347-353: Long sentence and requires adding references. References added.

Line 391: transfer [153,161]. Corrected

Line 400-401: Are these commercial brands? Is there any reference?References added.

Lines 406-410: Long sentence and requires adding references. References added.

Line 567: Describe all the table abbreviations in the table footer. Abbreviations described.

Reviewer 2 Report

The review of “Orchid Micropropagation Using Conventional Semi-solid and Temporary Immersion Systems: A review” for Plants MDPI.

The topic of manuscript fits within the scope of the journal Plants MDPI. The presented review is of interest from the point of view of reproduction and conservation of the biological diversity of rare species of orchids using modern methods of biotechnology. The current review highlights different aspects of in vitro orchid propagation using semi-solid and temporary immersion system and their benefits and drawbacks on rapid plant generation.

I have just a few remarks, which I give under author’s consideration:

1. Introduction

1.1. L 57-58: I do not recommend using the term "hormones", it is more correct to use the term "growth regulators".

1.2. L 66: Please check the used abbreviation "TIB", correct or explain this abbreviation by the first mention. This question arises since the abbreviation "TIS" was previously given.

2. Orchid Micropropagation Using Semi-solid Media

2.1. L 100: I do not recommend using the term "hormones", it is more correct to use the term "growth regulators".

2.2. L 122: Please move Figure 1 in the text of the manuscript after the first reference to this figure.

2.3. L 124: Please use one abbreviation for 6-benzylaminopurine in your article. Please check throughout the manuscript and unify the abbreviation used.

2.3. L 169-170: Please use unified concentration units (mg L-1 or μM) in the manuscript.

3. Orchid Micropropagation Using TIS

3.1. L 176-215: I suggest that lines 176-215 should be moved to a new section of the manuscript. For example: "2.2. Limitations of SS and liquid culture".

3.2. L 226-231: Please move Figure 2 and Figure 3 in the manuscript after the first reference to these figures.

3.2. L 259-261: In Figure 2 letters (c, d, e, f) are poorly distinguishable against the background of the pictures. Please put letters (a-f) below the corresponding photos.

Author Response

  1. Introduction

1.1. L 57-58: I do not recommend using the term "hormones", it is more correct to use the term "growth regulators". The term was replaced by "plant growth regulators".

1.2. L 66: Please check the used abbreviation "TIB", correct or explain this abbreviation by the first mention. This question arises since the abbreviation "TIS" was previously given. Abbreviation changed to "TIS".

  1. Orchid Micropropagation Using Semi-solid Media

2.1. L 100: I do not recommend using the term "hormones", it is more correct to use the term "growth regulators".The term was replaced by "plant growth regulators".

2.2. L 122: Please move Figure 1 in the text of the manuscript after the first reference to this figure. Figure has been moved.

2.3. L 124: Please use one abbreviation for 6-benzylaminopurine in your article. Please check throughout the manuscript and unify the abbreviation used. Abbreviation applied throughout the text.

2.3. L 169-170: Please use unified concentration units (mg L-1 or μM) in the manuscript. As the present manuscript is a review paper we  prefer to keep the  original units used by the cited authors. We would definitely keep all the measurement/ concentration units uniform if the manuscript is our research paper not a review paper. 

  1. Orchid Micropropagation Using TIS

3.1. L 176-215: I suggest that lines 176-215 should be moved to a new section of the manuscript. For example: "2.2. Limitations of SS and liquid culture". Lines moved and new section created as suggested.

3.2. L 226-231: Please move Figure 2 and Figure 3 in the manuscript after the first reference to these figures. Figures moved.

3.2. L 259-261: In Figure 2 letters (c, d, e, f) are poorly distinguishable against the background of the pictures. Please put letters (a-f) below the corresponding photos. Letters replaced for better visualization.
